In vitro feasibility of bovine and canine whole blood and commercially prepared canine packed red blood cells as a source of xenotransfusion in swine (Sus scrofa domestica)

Diaz Victoria 1
http://orcid.org/0000-0002-5260-2180 Schaefer Deanna M. W. 2
http://orcid.org/0000-0001-8722-4370 Mulon Pierre-Yves 3
Zhu Xiaojuan 4
http://orcid.org/0000-0002-4288-2262 Smith Joe 3
http://orcid.org/0000-0002-7329-683X Giori Luca 2
http://orcid.org/0000-0002-8344-8569 Hampton Chiara 3 champ14@utk.edu
1 College of Veterinary Medicine, University of Tennessee , Knoxville, Tennessee , United States
2 Department of Biomedical and Diagnostic Sciences, College of Veterinary Medicine, University of Tennessee , Knoxville, Tennessee , United States
3 Large Animal Clinical Sciences, University of Tennessee, College of Veterinary Medicine , Knoxville, Tennessee , United States
4 Office of Innovative Technologies, University of Tennessee—Knoxville , Knoxville, Tennessee , United States
Tomar Mahendra
Electronic publication date: 2025 May 9
Publication date: 2025
Volume: 13
Electronic Location ID: e19436
Received 2024 Nov 22; Accepted 2025 Apr 16
Copyright: © 2025 Diaz et al.
Copyright year: 2025
Copyright holder: Diaz et al.
License: This is an open access article distributed under the terms of the Creative Commons Attribution License, which permits unrestricted use, distribution, reproduction and adaptation in any medium and for any purpose provided that it is properly attributed. For attribution, the original author(s), title, publication source (PeerJ) and either DOI or URL of the article must be cited.
License URL: https://creativecommons.org/licenses/by/4.0/

Keywords: Blood, Bovine, Canine, Crossmatching, Dog, In vitro, Pig, Swine, Xenotransfusion

Funding: Biomedical and Diagnostic Sciences Department at the University of Tennessee, College of Veterinary Medicine This research was supported by the Faculty Educational Advancement and Research Fund made available by the Biomedical and Diagnostic Sciences Department at the University of Tennessee, College of Veterinary Medicine. The funders had no role in study design, data collection and analysis, decision to publish, or preparation of the manuscript.

==============================
Background: Since sourcing porcine blood donors for emergent transfusions to porcine patients is difficult, bovine or canine blood donors might represent alternative sources. The primary objective of this study was to determine the frequency of incompatible major (CMMa) and minor (CMMi) crossmatches by the standard saline agglutination tube method (SSA) between bovine whole blood (bWB) and whole blood from commercial pigs (pWB), and canine universal donor whole blood or commercially-prepared packed red blood cells (pRBCs) with whole blood from companion pigs. A secondary objective was determining the agreement between the reference method (SSA) and a quick slide (QS) method.

Methods: Blood was collected from 12 heifers, seven companion pigs, and eight commercial-cross pigs. A0 blood typing was performed for all porcine samples. Bovine blood was pooled into eight bags each containing three crossmatch-compatible individuals. Canine blood included whole blood from three canine blood donors (DEA 1.1, 5, 7 negative, and DEA 4 positive), and three bags each of DEA 1.1 negative and DEA 1.1 positive pRBCs. Crossmatch pairs were performed for bovine-to-porcine (n = 64) and canine-to-porcine (n = 63) samples. Incompatibility was defined as any agglutination or hemolysis on either CMMa and CMMi and reported separately. Complete incompatibility was defined as incompatibility of both CMMa and CMMi on the same pair. Kappa statistics tested the agreement between SSA and QS (significance at P < 0.05).

Results: For bWB and pWB, agglutination was observed in 9.4% of CMMa and 100% of CMMi via SSA. Incompatibility on CMMa of bWB was more frequent with porcine blood type “0” (P = 0.0107) than with type “A”, whereas porcine blood group had no effect on CMMi results. All canine-to-porcine CMMa were incompatible with SSA and showed hemolysis severe enough to prevent evaluation of agglutination. The accuracy of QS at detecting incompatibilities was 87.5% in CMMa and 98.4% in CMMi in bovine-to-porcine samples. Agreement between SSA and QS methods was fair (k = 0.36) for bovine-to-porcine CMMa but could not be calculated for CMMi due to lack of compatible matches. Because all canine-to-porcine CMMa were incompatible, the effects of the porcine blood group on incompatibility, accuracy of QS, and agreement between SSA and QS could not be calculated for CMMa. For CMMi, the agreement between tests was poor (k = 0).

Discussion: When a xenotransfusion to a pig is indicated, bWB appears to be suitable based on in vitro CMMa testing, whereas canine blood products are contraindicated for in vivo administration to swine based on absolute CMMa incompatibility and incidence of hemolysis. In vivo studies are needed to elucidate the clinical significance of CMMi incompatibilities. Based on these results, QS cannot be accurately used as a surrogate of SSA in pretransfusion testing for porcine patients due to the increased risk of false compatible results as QS can only be identified as agglutination, not hemolysis.

INTRODUCTION

Over the last three decades, pigs have become established companion pets, in addition to remaining a fundamental food source and one of the most prominent experimental pre-clinical models for translational research (Swindle, 2007). This population growth unfortunately has not been accompanied by a proportional increase in evidence-based medical practices dedicated to this species, and clinical decisions are often made based on anecdotal knowledge, or by applying guidelines and practices established for other species. Therefore, species-specific evidence-based information is needed to meet the clinical standards of care received by canine, feline, and equine patients.

Transfusions of blood products are administered with the intent to replace blood components, either red blood cells (RBCs), plasma, or coagulation factors. When whole blood (WB) or packed red blood cells (pRBCs) are selected for administration to a recipient, an additional aim is to increase the patient’s oxygen carrying capacity, which may have been compromised by conditions such as chronic or acute loss of blood, hemolysis, ineffective erythropoiesis, immune-mediated hemolytic anemia, chronic inflammation, and/or neoplasia (Kumar, 2017; Aravindh & Jacob, 2021). In swine, metabolic conditions such as mineral and vitamin deficiencies (e.g., iron, copper, cobalt, B3, B5, B6, B9, and B12), toxicities (e.g., mycotoxins and rodenticides), infectious diseases (e.g., Mycoplasma suis), parasitism (e.g., Ascaris suum and others), and other gastrointestinal pathologies (e.g., esophageal ulcers, gastric ulcers, hemorrhagic ileitis, proliferative enteritis) may cause anemia and decreased oxygen carrying capacity (Clark & Coffer, 2008). Elective surgical procedures such as ovariohysterectomy and castration can be another cause of hemorrhagic anemia in pigs and are being performed with increasing frequency as the population of pigs presenting to veterinary practitioners increases. In a retrospective study, surgical procedures involving the neoplastic reproductive tract have been shown to yield high percentages of hemorrhage, which in some cases has led to a patient’s cardiovascular collapse and death (Cypher et al., 2017; McOnie et al., 2021).

Although significant efforts have been made to successfully use pigs as a xenotransplantation source for humans (Cooper, 2003; Roux, Saï & Deschamps, 2007a, 2007b; Wang et al., 2016), evidence-based clinical guidelines for transfusion of blood products in pigs are non-existent. In fact, this species consistently fails to appear on review articles published on the matter due to a lack of evidence-based recommendations (Credille & Epstein, 2016; Kumar, 2017). The reasons for this lack of evidence-based information are multiple. Firstly, pigs have only recently acquired “pet” societal status, encouraging the owner’s financial commitment. Secondly, ideal crossmatching techniques in pigs have not been established, hindering blood donor selection when treating an anemic pig with transfusions. Furthermore, the significance of the incompatibility of porcine A0 blood groups in promoting major transfusion reactions has not been established. Finally, there are no commercially available blood products for pigs (Credille & Epstein, 2016) and blood collection from a porcine donor is notoriously difficult due to the hardship in gaining reliable venous access in this species. The process is invasive due to the need for general anesthesia and potential surgical cut-downs to harvest blood (Elane et al., 2024). As seen in our veterinary practice, lack of access to blood to restore circulating volume and oxygen carrying capacity may contribute to perioperative morbidity and mortality in this species (Cypher et al., 2017; McOnie et al., 2021).

Xenotransfusion is the practice of administering blood products harvested from a subject of one species to a subject of another (Roux, Saï & Deschamps, 2007a). This practice has been investigated in several species as a mean of providing a one-time emergency transfusion based on the assumption that the recipient would not have been previously exposed to blood from the donor species, and therefore it would lack antibodies against those foreign (xeno-) antibodies (Bovens & Gruffydd-Jones, 2013; Euler et al., 2016; Oron et al., 2017; Buck et al., 2018; Le Gal, Thomas & Humm, 2020; Smith et al., 2021; James et al., 2022). For example, the current evidence on using dogs as donors for feline recipients is that “canine blood can be administered to cats in genuine emergency situations when no other options exist, provided the cat has not received dog blood previously” (Caroline, 2016). Due to the current difficulties in sourcing swine blood, we are evaluating xenotransfusions to pigs to verify if this practice could represent a viable clinical practice with the potential to provide therapeutic benefit in the emergency settings due to its ability to temporarily stabilize the patient’s cardiovascular system. This would allow additional time for essential diagnostic and surgical procedures to be performed, and a more suitable species-specific donor to be identified (Bovens & Gruffydd-Jones, 2013; Euler et al., 2016).

Bovine blood has been investigated as a potential source of blood to produce blood products for humans, due to its low chance of adverse reaction to the recipient’s blood (Johnstone et al., 2004) and bovine hemoglobin-based oxygen carrier products exist for therapeutic tissue oxygenation (Gupta, 2019). Furthermore, cattle have easily accessible blood vessels, which makes the collection of large amounts of WB logistically and technically easier than harvesting blood from a donor pig. The volume of blood that can be safely collected from cattle is also greater than in pigs, allowing the administration of larger volumes during transfusion to more effectively raise the packed cell volume regardless of the size of the recipient pig. However, since some small animal veterinary practices may occasionally treat pet pigs, blood products available to small animal practitioners such as canine whole blood (cWB) and canine packed red blood cells (cPRBCs) are also of interest as xenotransfusion sources. To our knowledge, there are currently no veterinary studies assessing the feasibility of xenotransfusion of either WB from bovine donors (bWB) or of canine blood products to porcine recipients. Crossmatching is a serological method that, along with blood typing, constitutes the base of pretransfusion testing (Sidhu & Shah, 2020). The purpose of these tests is to assess the potential for adverse transfusion reaction by evaluating for agglutination and/or hemolysis between the donor’s and recipient’s blood. Two types of crossmatches are typically performed. The major crossmatch, which tests whether antibodies in the serum of the recipient cause agglutination or hemolysis of the donor RBCs, is the most clinically relevant test, and an incompatibility contraindicates use of that donor (Wardrop, 2022). Incompatibilities in the minor crossmatch, which tests donor’s serum against recipient’s RBCs, usually do not cause life-threatening transfusion reactions (Bovens & Gruffydd-Jones, 2013).

Therefore, the primary aim of this study was to determine the frequency and degree of incompatible crossmatch (minor and major) reactions via a standard tube method (SSA) and quick slide technique (QS) using (1) bWB and porcine whole blood (pWB), and (2) different types of canine blood products and companion swine whole blood (cpWB), where the types of canine blood products evaluated included cWB (DEA 1.1 negative), cPRBCs (DEA 1.1 negative), and cPRBCs (DEA 1.1 positive). The null hypotheses were that crossmatching assessed by SSA and QS would yield fewer than 40% incompatible reactions (micro- and macro-agglutination or hemolysis) for bWB (donor) and pWB (recipient) crossmatches or for canine blood products (donor) and cpWB (recipient) crossmatches. A secondary aim of this study was to determine the level of agreement between the results obtained by SSA and those obtained via QS to verify if the latter would constitute a quicker and easier alternative to the classic crossmatching procedure via SSA to be used in the emergency setting. The null hypothesis was that the value of the κ statistics between the results produced by SSA and QS would be higher than 0.4.

Materials and Methods

Design

This experiment was designed as a prospective observational in vitro study conducted on a total of 127 crossmatching paired tests.

Animals

This study was approved by the Institutional Animal Care and Use Committee at the University of Tennessee (Protocol # 2965-0323). Client consent was obtained for blood sampling from all porcine subjects prior to collection. Due to the lack of similar studies and existing data on this topic, there was insufficient information to calculate a sample size via power analysis. Based on the currently known distribution of blood groups in pigs (Smith et al., 2006; Hampton et al., 2023), a sample size of four pigs per major A0 blood group system was deemed to be needed (a total of eight commercial pigs and eight pet pigs, acting as the “recipients”) to be representative of the blood group population. Inclusion criteria for swine were age (greater than 5 weeks) and weight (greater than 1 kg). There are 11 major blood group systems in cattle, and more than 100 antigens are currently identified. Some groups are particularly rare (M, R, T, TF, Z) (Rocha et al., 1998). Blood typing is no longer performed in cattle by commercial laboratories due to the complexity of the antigenic system. Therefore, the sample size of bovine donors was based on previous references (Dell, Holleran & Ramakrishnan, 2002) which indicated that assuming that b was 0.05 with the occurrence of bovine blood types being 4/11 (e.g., A, B, C, L) and maximize antigen exposure (Rocha et al., 1998), a minimum of five bags of pooled blood from at least 11 cattle would be needed. Inclusion criteria for cattle and dogs were normal physical examination prior to blood sampling.

In vitro “donor” blood samples

Bovine whole blood was used as a “donor” sample for crossmatching with samples from commercial-bred pigs. Approximately 10 mL of blood was collected from the jugular or coccygeal veins of 12 Holstein heifers and tested for compatibility via a complete crossmatching technique. Compatibility of blood from the three heifers pooled in the same bag was verified prior to commencing the porcine study and prior to the pooling of bovine blood. The contribution of each heifer was limited to two pooled bags, to allow for identification of the subject that would have caused a potential reaction of incompatibility (e.g., if both bags in which the blood from the same subject reacted with porcine blood). After a 2-month period, an additional 60 mL of blood was collected from the same compatible heifers and pooled into eight bags (feline blood collection bag, 100 mL with ACD J0520R Jorvet) with added acid citrate dextrose (ACD) at an optimal ratio (12.5%) (Orr et al., 2021) and stored at 4 °C for a maximum of 7 days before usage for crossmatch. The combination scheme of the pooled blood is exemplified in Table 1.

Table 1 Scheme of bovine blood pooling to increase antigen exposure with porcine blood samples during in vitro crossmatching.

Pool bag code	Subject combination	
A	B1/B2/B3	
B	B1/B4/B5	
C	B2/B6/B7	
D	B3/B8/B9	
E	B4/B10/B11	
F	B5/B6/B12	
G	B7/B9/B11	
H	B8/B10/B12	
Note:

B, Bovine; 1–12, Subject.

The “donor” samples for the crossmatch procedures to be performed against blood samples from companion pigs included three canine whole blood (cWB) universal donor samples (DEA 1.1 negative, DEA 4 positive, DEA 5 negative, DEA 7 negative, n = 3), and commercially purchased canine universal donor (DEA 1.1 negative, DEA 4 positive, DEA 5 negative, DEA 7 negative, n = 3) and DEA 1.1 positive (n = 3) packed red blood cells (cPRBCs) bags. Commercial canine products were purchased from HemoSolutions (Colorado Springs, CO, USA). Canine whole blood (10 mL) was sampled from three healthy dogs (two desexed male and one female) which had previously been blood typed and profiled with an IDEXX Blood Typing Complete Panel. Samples were stored at 4 °C prior to performing crossmatchings. Bags of cPRBCs were purchased and stored at 4 °C for a maximum of 15 days before usage for the crossmatching procedure.

In vitro “recipient” blood samples

Blood samples (10 mL) from eight commercial pigs (six barrows and two sows, weight 255.9 ± 72.9 kg, age 7.1 ± 2.7 years) and seven companion pigs (one boar, four sows, two gilts, weight 69.6 ± 30.9 kg, age 8.2 ± 5.8 years) were opportunistically collected under sedation or general anesthesia from various collection sites (vena cava, jugular, tarsal, auricular veins) and placed in EDTA (Blood Collection Tube, EDTA 7.5%; Lavender, Cardinal Health) and serum collection tubes (Blood Collection Tube, Serum, Red Lid, No Additives; Cardinal Health). Blood typing was performed via a clinically validated method (EldonCard 2511; Eldon Biological A/S, Gentofte, Denmark) (Hampton et al., 2023; Hampton, Zhu & Giori, 2023) according to the A0 blood group system, and used as the “recipient” samples for the crossmatch procedure. Four blood type “A” commercial and companion pigs, and four blood type “0” commercial and companion pigs were included in the study. Health status was not an exclusion criterion based on previous references (Hampton, Zhu & Giori, 2023). Three mL of WB samples were placed in labeled round-bottom borosilicate tubes (Fisherbrand round bottom disposable tube; Thermo Fisher Scientific) and centrifuged at 2,600 g for 3 min, and plasma (at least 750 mL) was transferred to a new labeled tube.

Washed red blood cell procedure

All red blood cell (RBC) samples were washed and diluted to a 4% suspension. Fifty µL of the donor’s and recipient’s RBCs were placed in a round bottom tube and filled with approximately 2 mL of phosphate buffered saline (PBS; Thermo Fisher Chemicals). The samples were centrifuged at 1,000 g for 1 min and saline aspirated and discarded. This washing procedure was repeated 2 more times with a final 1.2 mL of PBS added to create the final 4% suspension.

Standard saline tube agglutination

The following procedure was standardized from Newman (2014). A single blood sample was collected from each “recipient”. Each sample from commercial pigs was crossmatched with eight pooled bags of bovine blood (n = 64 pairs). Each sample from companion pigs was crossmatched with three samples of cWB (n = 21 pairs), three samples of cPRBCs DEA 1.1 positive (n = 21 pairs), and three samples of cPRBCs DEA 1.1 negative (n = 21 pairs). Complete crossmatching was performed on all donors and recipient samples, including one auto-control per sample within 4 h of blood collection from the “recipients”. Sample preparation included making a 4% RBC suspension by using calibrated pipettes as previously described. The CMMa was performed using 100 µL of recipient serum and 50 µL of donor’s 4% RBCs. The CMMi was performed using 100 µL donor plasma and 50 µL recipient 4% RBCs. The auto-control was performed using 100 µL recipient serum and 50 µL recipient 4% RBCs. Some control samples (QS: porcine plasma + porcine pRBC; SSA: porcine serum + washed porcine pRBC) were performed with the use of with rabbit or guinea pig complement. All samples were incubated for 30 min at 37 °C. After incubation, the tubes were centrifuged at 1,000 g for 30 s (LW Scientific, Zip-IQ TT Test Tube Centrifuge). Agglutination was first evaluated macroscopically based on a graded scale (Table 2) (Guzman, Streeter & Malandra, 2016), and if none was detected, it was evaluated microscopically. The presence of microscopic agglutination was reported as binary (positive or negative). Samples were evaluated for hemolysis in the supernatant and compared to the auto-control with a standardized scoring system shown in Fig. 1 and Table 3, Newman (2014). Incompatibility was defined as any macroscopic or microscopic agglutination or hemolysis. All crossmatchings, including macroscopic and microscopic evaluations were performed by the same individual (VMD), who was extensively trained prior to the commencement of the study. Examples of SSA results are shown in Figs. 2A and 2B.

Table 2 Scoring scale for agglutination grading (Guzman, Streeter & Malandra, 2016).

Grade	Description	
0	No agglutination visible microscopically or macroscopically.	
1+	No macroscopic agglutination, but weak or transient microscopic adherence of RBCs where there are groups of 2-3 cells that appear loosely aggregated and may be difficult to distinguish from rouleaux. Dilute the sample further by mixing equal parts of sample and saline and reevaluate to see if the associations disperse.	
2+	No macroscopic agglutination, but small microscopic agglutinates present (4–10 RBCs per agglutinate).	
3+	No macroscopic agglutination, but at least 1 large microscopic agglutinate (>10 RBCs per agglutinate).	
4+	Macroscopic (grossly visible) agglutination.	

Figure 1 Grading scale for evaluation of hemolysis as described in Table 3.

The degree of hemolysis increases from left to right, with 1 representing slight hemolysis and 6 representing marked/complete hemolysis. Photo courtesy of Dr. Deanna Schaefer.

Table 3 Scoring scale for hemolysis grading (Newman, 2014).

Grade	Description	
0	No visually detectable hemolysis	
1+	Slight hemolysis
(tube 1 in Fig. 1)	
2+	Moderate hemolysis
(tubes 2–3 in Fig. 1)	
3+	Marked hemolysis
(tubes 4–6 in Fig. 1)	

Figure 2 Examples of standard saline agglutination results for compatible major crossmatching (A) and incompatible minor crossmatching (B).

Quick slide procedure

Complete crossmatching via QS was performed on all donors (bovine and canine) and porcine samples with one auto-control for every porcine sample. The CMMa was done by mixing two drops of recipient plasma with one drop of donor pRBCs on a microscope slide. The CMMi was done by mixing two drops of donor plasma with one drop of recipient pRBCs on the microscope slide. The slides were rotated for 2 min to allow thorough mixing of plasma and RBCs while assessing for macroscopic agglutination. If CMMa and CMMi slides were free from macroscopic agglutination, they were evaluated for microscopic agglutination by performing a saline dilution test to distinguish agglutination from rouleaux. A microscopic dilution of the auto-control was also performed for comparison. In brief, four drops of saline from a transfer pipette were placed on a microscope slide with a small amount of the slide mixture from a pipette tip to create an approximate 1:50 dilution of RBCs. The slide was gently rotated to mix blood and saline. A coverslip was placed over the sample and evaluated for microscopic agglutination. Examples of QS results are shown in Figs. 3A and 3B.

Figure 3 Examples of quick slide method results for compatible major crossmatching (A) and incompatible minor crossmatching (B).

Statistical analysis

Results from crossmatching and microscopic evaluations were recorded manually and then transcribed and stored in a commercially available spreadsheet (Excel, Microsoft, Redmond, WA, USA). Incidence of compatibility was reported in percentage for complete crossmatching, CMMa, and CMMa for both techniques. Mann-Whitney U test was used to determine the difference between porcine blood groups in terms of compatibility with bovine blood. The relative risk of compatibility was calculated as follows for porcine recipients of blood groups “A” and “0” with bovine donors:

RelativeRiskofCompatibility=%ofcompatibleCMMaofbWBwithpCMMaofgroup"A"%ofcompatibleCMMaofbWBwithpCMMaofgroup"0"

The accuracy of QS was calculated via the following formula:

[(#CM−#ofdiscordantCM)#CM]∗100

Kappa statistics was used to test the level of agreement between SSA and QS. Kappa statistics were categorized as poor (0–0.20), fair (0.21–0.40), moderate (0.41–0.60), good (0.61–0.8), or very good (0.81–1.00). Parametric and non-parametric data are reported as appropriate. Significance was set at P < 0.05. MedCalc® Statistical Software version 22.009 was used for all the statistical analysis (Ostend, Belgium; https://www.medcalc.org; 2023).

Results

Twelve healthy Holstein heifers aged 237 ± 32 days, three dogs (two male castrated and one spayed female; 3.9 ± 2.3 year old), eight commercial pigs (six male castrated and two intact females; 7.1 ± 2.7 year old; 255.8 ± 72.9 kg), and seven companion pigs (one male castrated, four intact female, three female spayed; 8.2 ± 5.7 year old; 69.6 ± 30.9 kg) were enrolled in the study. Breeds are specified in Table 4. Collection of samples from companion pigs was stopped after seven pigs since all samples were highly incompatible.

Table 4 Summary of breeds and number of subjects enrolled in the present study.

Species	Number of subjects	Breed	
Bovine	12	Holstein	
Canine	1	Bulldog mix	
1	Husky mix	
1	Labrador retriever	
Porcine (Commercial)	3	Large white	
2	Wild boar	
2	Spotted	
1	Yorkshire cross	
Porcine (Companion)	3	Vietnamese potbellied pig	
2	American mini pig	
2	Kune kune	

Bovine-to-porcine whole blood crossmatches

Sixty-four pairs of crossmatchings were performed on bWB and pWB samples via SSA and QS. Briefly, agglutination was observed in 9.4% of CMMa, and in 100% of CMMi via SSA of bWB and pWB, and in 9.4% and 98.4% via QS on the same samples. Therefore, in vitro compatibility between bWB and pWB based on CMMa was 90.6%. Some control samples resulted in agglutination and hemolysis despite being from the same patient when complement was added to the crossmatching procedure. When using saline in place of the complement, there were no agglutination or hemolysis noted.

Canine-to-porcine whole blood and pRBCs crossmatches

Sixty-three pairs of crossmatchings were performed on cWB and cpWB samples via SSA and QS. All canine-to-porcine CMMa on SSA and QS were incompatible with all tested canine products, independent of the DEA antigen profile. In the canine-to-porcine crossmatchings, hemolysis was severe enough to prevent the evaluation of agglutination. 2/6 auto-control were scored as a 1+ on SSA for agglutination, and 0/6 presented hemolysis.

Effect of porcine blood type on compatibility

The compatibility of bWB with porcine blood type “A” on CMMa was significantly greater than that of pBW type “0” (P = 0.0107), with a relative risk of compatibility of 1.23. Porcine blood group had no effect on CMMi results. The effects of the porcine blood group on incompatibility could not be calculated for canine-to-porcine CMMa due to the absolute lack of compatible matches.

Performance of quick slide method

Incompatibility reactions based on agglutination and hemolysis on SSA were compared to incompatibility based on agglutination only on QS due to the fact that this technique is unable to detect hemolysis. The accuracy of QS was 87.5% for detection of incompatibilities in CMMa and 98.4% in CMMi in bovine-to-porcine samples. Agreement between SSA and QS methods was fair (κ = 0.36) for bovine-to-porcine CMMa but could not be calculated for CMMi due to lack of compatible matches. The accuracy of QS and agreement between SSA and QS could not be calculated for canine-to-porcine CMMa due to the absolute lack of compatible matches. For CMMi, the agreement between tests was poor (κ = 0). The QS control samples contained individual porcine plasma and porcine pRBC. The QS still resulted in a positive microscopic agglutination (1+). However, when performed on the SSA, there were no signs of reaction (hemolysis or agglutination), once again showing that the SSA is the more reliable testing method.

Discussion

This is the first study investigating the in vitro compatibility of porcine blood with blood products from two other species, bovine and canine. Our findings suggest that due to the high frequency of compatibility on major crossmatching between bovine donors and porcine recipients, the practice of xenotransfusion between these species may be supported. Based on the relative risk of compatibility found in this study, a xenotransfusion with bWB to a porcine patient with type “A” blood would be 1.23 times more likely to be compatible than a porcine patient with type “0” blood. On the contrary, due to the high incidence of hemolysis on both major and minor crossmatching between canine products and porcine samples, we strongly discourage in vivo xenotransfusions involving a canine donor and a porcine recipient.

The aim of the CMMa procedure is to evaluate for antibodies in the recipient’s plasma that react with the donor’s RBCs (Sidhu & Shah, 2020; Wardrop, 2022). Incompatibility on CMMa is considered a contraindication for the administration of blood products from one individual to another, no matter the species. Even though this concept has been challenged by some recent findings, that is, cats being given type specific blood with and without crossmatching (reaction criteria being 1+ agglutination and hemolysis of 1−2+) (Sylvane et al., 2018), the current recommendation is to use plasma and RBCs for canine-to-feline xenotransfusions that are at least compatible on a CMMa, since this type of crossmatch best mimics host immunity (Weltman, Fletcher & Rogers, 2014; Wardrop, 2022). Bovine blood has already been evaluated as a potential transfusion source for humans (Johnstone et al., 2004). In a recent in vitro crossmatching study, bovine RBCs were commonly compatible on the major crossmatch with canine plasma (Prado et al., 2024). Similarly, in our study, the majority of crossmatches between bovine RBCs and porcine plasma were also compatible, providing more support for the use of bovine blood in xenotransfusions. Because there are a large number of bovine blood groups, it is statistically probable that the bovine donors used in our study expressed a variety of red cell antigens, yet the majority of crossmatches with porcine plasma were still compatible, suggesting that it is unlikely that pigs produce naturally-occurring antibodies to bovine red cell antigens. However, the minor crossmatch pairs between porcine RBCs and bovine plasma were all incompatible by primarily showing hemolysis, supporting that cows do have naturally-occurring antibodies to both type A and type 0 pig blood. Heifers were used to prevent any potential allo-antibodies that could be circulating from prior sensitization to a fetus with a different blood type. We did not evaluate whether or not there would be any incompatibility with the use of cows vs. heifers with any in vitro samples. Multiparous cows were the main donors for xenotransfusions to goats, and there were no extreme reactions (Smith et al., 2021).

The aim of the CMMi procedure is to verify the compatibility of the donor’s plasma and the recipient’s RBCs (Sidhu & Shah, 2020; Wardrop, 2022). Most paired bovine-to-porcine samples were incompatible on CMMi, which would test for the presence of anti-porcine antibodies in transfusion-naïve bovine donor plasma. Incompatibility on CMMi is considered less clinically significant than the CMMa unless transfusing blood products rich in plasma content (Tocci & Ewing, 2009), as they are less likely to cause a transfusion reaction, since the donor plasma would be diluted in vivo, leading to less exposure of the recipient RBCs to foreign plasma (Bovens & Gruffydd-Jones, 2013). Supporting this, two studies of canine to feline xenotransfusions showed that cases with incompatible minor crossmatches had no clinically detectable adverse effects despite administration of transfusion (Bovens & Gruffydd-Jones, 2013; Le Gal, Thomas & Humm, 2020). However, another study showed that the average lifespan of canine erythrocytes transfused into cats was very short, less than 4 days (Euler et al., 2016). Therefore, based on the minor crossmatch incompatibilities found in this study, using solutions of bovine pRBCs where some of the plasma is removed might reduce the risk of transfusion reactions in the porcine recipient. Further studies would be needed to determine if the circulating lifespan of bovine erythrocytes in xenotransfusion is similarly short, but the results highlight that xenotransfusion is likely to be most useful for short-term management of critical patients, such as those with acute blood loss where compatible same-species donor blood is not available (Bovens & Gruffydd-Jones, 2013).

In contrast to our results using bovine blood, all major crossmatch results between canine erythrocytes and porcine plasma were incompatible. Additionally, many of the minor crossmatches were also incompatible, typically showing both hemolysis and agglutination. This supports the idea that both pigs and dogs have naturally-occurring antibodies to red cell antigens of other species. For this reason, administration of canine blood to swine patients is not recommended, even in the immediate need for life-saving xenotransfusion. A possible explanation could reside in the fact that cattle and domestic pigs fall within the same taxonomic order (Artiodactyla), whereas dogs in the order Carnivora. Therefore, it would be fair to speculate that bovine RBCs would be more similar to porcine RBCs or at least evolved from those of a more recent common ancestor.

In applicable clinical conditions as deemed by the clinician and patient presentation, xenotransfusions have been performed between a wide variety of species. In these situations, the nearest similar species has usually been the donor of choice. While the resources are slim for species beyond the major veterinary species (canine, feline, and equine) for transfusion therapy, there are cases of in vivo xenotransfusions in lesser studied species. The majority of xenotransfusions in large animals are with bovine donors (Brown & Vap, 2012; Buck et al., 2018; Smith et al., 2021; James et al., 2022). Despite the lack of literature on porcine xenotransfusions, there is enough evidence from other species studies that even with an incompatible result (in this case the minor), a xenotransfusion can still be life-saving. Recipient sensitization occurs in rabbits similar to other mammals (Bovens & Gruffydd-Jones, 2013; Kisielewicz & Self, 2014; Kumar, 2017; Dannemiller et al., 2024). In vivo studies would be needed to evaluate whether simply having crossmatch compatible blood is effective in increasing the porcine recipient’s PCV post-transfusion (Weltman, Fletcher & Rogers, 2014).

Our results support that the quick slide method should not be used as a surrogate of SSA in porcine patients for pre-transfusion testing due to the increased risk of false compatible results between donor and recipient, although it may perform better in other species in detecting agglutination. Regardless of species, the slide agglutination method doesn’t test for hemolysis which is the most important pre-transfusion clinical in vitro finding. Statistically, the test is not capable of detecting as many complete incompatibilities as the reference method. There is then a risk of false compatibility of the crossmatchings in an incompatible donor/recipient pair if only QS is used. Rather, a complete crossmatch procedure by saline tube agglutination or using a crossmatch kit is recommended.

Several limitations deserve mention in this discussion. Causes of incompatible crossmatch could be due to patient or donor unit factors, but also to technical or clerical errors as previously reported by Sidhu and Shah in 2020 and potential and inherent errors may have affected our results. However, we followed rigorous standard operating procedures created ad hoc for this study to minimize these errors. The detection of hemolytic reaction in large animal species usually requires the addition of complement, in this case commercially available guinea pig or rabbit serum (Clark & Coffer, 2008; Wardrop, 2022), since incompatibilities in these species may sometimes result only in hemolysis without agglutination. In our study, the addition of guinea pig or rabbit complements hemolyzed all crossmatchings pairs, indicating that the animals that were the source of the complement had antibodies against pig erythrocytes. These cross-reacting antibodies can be removed from the complement reagent through an absorption procedure (Brown & Vap, 2012). However, the addition of complement with this absorption procedure is expensive and time-consuming and may not be feasible for many laboratories performing crossmatching procedures. Therefore, we acknowledge that hemolysis may have been under-detected in this study. This was easily detected when the SSA was performed with the canine blood, even without the addition of complement. Another limitation is the presence of positive results of the auto-control which may affect our ability to accurately determine whether incompatibility is a normal attribute compared to the control or an actual positive result. This phenomenon only occurred with the quick slide method and therefore, has limited clinical significance as this method has been shown to be unreliable in the current study. Finally, testing for bovine blood groups is clinically not available. Therefore, our study cannot point to blood groups or antigens that were responsible for incompatibility. This limitation was handled by pooling bovine blood groups to increase antigen exposure to porcine blood. To our point, we were able to isolate two individual donors that had at least one incompatible result (Appendix 1). However, having a variety of individuals to do a crossmatch would be ideal in order to find a compatible donor for a transfusion. Finally, the small sample size of subjects used in this study may have affected its results. However, we believe the sample size calculated via power analysis was adequate to detect significance.

Conclusions

This study contributes to the current body of knowledge in the field of xenotransfusion. Based on these in vitro results, for any clinical scenario in which a blood product transfusion is indicated, bovine whole blood holds potential as a donor source for swine and appears to be suitable based on in vitro pre-transfusion testing, whereas canine blood products appear to cause severe hemolysis and are therefore considered contraindicated for administration to pigs. In vivo studies are needed to elucidate the clinical significance of the incompatibility of major and minor crossmatch in pigs. In order to improve the compatibility of bovine-to-porcine xenotransfusions based on CMMi, bovine pRBCs may be a suitable blood product for administration to swine recipients.

Supplemental Information

Supplemental Information 1 Individual results for bovine-porcine and canine-porcine crossmatching pairings.

Supplemental Information 2 ARRIVE Checklist.

The authors would like to thank the staff of the Large Animal Hospital for their help and support in collecting blood samples, particularly Ms. Katie Houser. We’d also like to thank Mr. Clay Kesterson and the Little River Dairy for caring for and maintaining the bovine herd. We would also like to extend our gratitude to the owners and refuges that have kindly enrolled their pigs in our study, and the owners of the dogs that acted as blood donors. The authors declare that there were no conflicts of interest.

Additional Information and Declarations

Competing Interests

The authors declare that they have no competing interests.

Author Contributions

Victoria Diaz performed the experiments, prepared figures and/or tables, authored or reviewed drafts of the article, and approved the final draft.

Deanna M. W. Schaefer conceived and designed the experiments, performed the experiments, prepared figures and/or tables, authored or reviewed drafts of the article, and approved the final draft.

Pierre-Yves Mulon conceived and designed the experiments, authored or reviewed drafts of the article, and approved the final draft.

Xiaojuan Zhu analyzed the data, authored or reviewed drafts of the article, and approved the final draft.

Joe Smith performed the experiments, authored or reviewed drafts of the article, and approved the final draft.

Luca Giori performed the experiments, authored or reviewed drafts of the article, and approved the final draft.

Chiara Hampton conceived and designed the experiments, performed the experiments, analyzed the data, prepared figures and/or tables, authored or reviewed drafts of the article, and approved the final draft.

Animal Ethics

The following information was supplied relating to ethical approvals (i.e., approving body and any reference numbers):

The Institutional Animal Care and Use Committee of the University of Tennessee provided full approval for this research (2965-0323).

Data Availability

The following information was supplied regarding data availability:

The raw data is available in the Supplemental File.

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
