# Peer review of "In vitro feasibility of bovine and canine whole blood and commercially prepared canine packed red blood cells as a source of xenotransfusion in swine (Sus scrofa domestica)"

_PeerJ, doi:10.7717/peerj.19436_

## Round 0.1 · original submission · Major Revisions

Please address the comments from the reviewers.

·

Basic reporting

The article has minor improvement to bemade. Some grammar error (attachd file).

Experimental design

No comment

Validity of the findings

The findings are interesting, however authors should write a discussion about the low number of animal tested, the influence of breed (specially canine and porcine) on results. Sample size for commercial pig and cattle is small, as reseachers could have plenty of animals from pigs/cattle farms. In the case of pet pigs, there was stated 8 pigs but only 7 are reported in table 4 with three breeds and two anima each. Then again, it might be useful to disucuss the results in terms of these issues.

Reviewer 2 ·

Basic reporting

Clear and professional English used throughout
Relevant results anwers to hypotheses

Experimental design

Suitable with aims and scope of the journal

Methods were descrided clear and data were enough for analysis

Validity of the findings

Meaningfull data and results for application of transfution

Reviewer 3 ·

Basic reporting

Thank you for your efforts to advance our knowledge of transfusion medicine in domestic porcine patients. This in vitro experimental pre-transfusion study is thoughtful and clinically applicable with the bonus of providing a method comparison between a conventional tube crossmatch and slide agglutination testing. However, the manuscript would benefit from several revisions that would improve its clarity and better communicate its methodology. Please see my specific comments below.

Experimental design

See below

Validity of the findings

See below

Additional comments

Manuscript: Please double-check that all "in vitro" and "in vivo" are italicized appropriately
Line 25: Consider revising the sentence to, "Since sourcing porcine blood donors for emergent transfusions to porcine patients is difficult, bovine or canine blood donors might represent a potential alternative."
Line 57-60: Please revise the sentence to, "…QS cannot be accurately used as a surrogate of SSA in pretransfusion testing for porcine patients due to the increased risk…"
Line 72-74: Please revise the end of the sentence as follows, "…clinical standards of care received by canine, feline, and equine patients."
Line 75-76: Please clarify "cells" to "red blood cells" or "RBCs"
Line 77: Please change "PRBCs" to "pRBCs" for consistency.
Line 75-107: The content of this paragraph is great, but it's a bit unwieldy in length. Consider splitting it into two before Lines 91-94 ("Although significant efforts have…")
Line 108-144: Similarly, consider splitting this paragraph into two prior to Lines 123-126 ("Bovine blood has been investigated…")
Line 142-144: This is incorrectly worded. Minor crossmatch detects antibodies in the DONOR's serum to the RECIPIENT'S red blood cells.
Line 174-175: Please change "and (weight greater than…" to "and weight (greater than…"
Line 205-209: Please clarify which antigens were absent/present for "DEA+" pRBCs and from what commercial source they were purchased
Line 213-216: Please clarify the abbreviation "CMs" as it is the first time it shows up in the body of the manuscript. Since you all have already introduced "CMMa" and "CMMi," I would recommend using those abbreviations or simply writing out "crossmatches" to avoid EUAs (extra unnecessary abbreviations).
Section Standard Saline Tube Agglutination:
1) Given the variation between protocols, please identify the reference or institution-specific protocol on which your standard saline agglutination tube crossmatches are based.
2) Did you use a calibrated pipette to measure the donor and recipient's serum and 4% RBC suspension?
3) I may have missed it, but please identify the type/model of the centrifuge used for the crossmatches.
4) Why did the authors decide not to evaluate hemolysis and agglutination before and after incubation?
Line 293-299: This information may be better suited in the Materials & Methods than the Results. me
Line 315-318: Please change "greater" to "significantly greater." I would also suggest that the authors calculate the relative risk of incompatibility for bovine WB with porcine blood type "A" or "0." See for more details: https://www.ncbi.nlm.nih.gov/books/NBK430824/.
Line 334-340: Great succinct and clinically applicable recap
Figure 1: Please identify who—or which source—provided this photo (e.g., Photo courtesy of…")
Line 350-352: This citation doesn't correspond to a listed reference. Please add this citation to your references and double-check all sources cited are listed.
Line 380-384: Remove "However,"
Line 387-389: Perhaps RBC antigens are more closely conserved (or at least more similar) amongst artiodactyls and more disparate between artiodactyls and carnivores?
Line 401-403: Consider re-wording "is effective on post-transfusion significantly improved PCV" to "is effective in increasing the porcine recipient's PCV post-transfusion"
Line 404-406: Please clarify that the slide agglutination method should not be used as a surrogate for porcine patients as it may perform better in other species to detect agglutination. It's also worth noting in your Discussion that regardless of species, the slide agglutination method doesn't test for hemolysis which is the more clinically concerning in vitro pre-transfusion finding.
Line 445: This sentence is somewhat anticlimactic. Maybe delete it or revise it?

·

Basic reporting

a. The English language used is in correct grammatical form with easy-to-understand sentences
b. The topic is well referenced with majority of the references cited from the last 15 years and a good number of references of the last five years i.e. after 2020 year
c. The standards of PeerJ are met and there is clarity in scientific expression
d. The tables are numbered correctly, and the data is explained well in the text of the manuscript
e. The following points may be addressed
i) Line 93 – Deschamps 2007b – is missing in Reference section
ii) Line 114 – J. S. Smith et al – remove initials while citing in text
iii) Line 172, 226, 227, 230, 364, 396 – remove initials from cited references
iv) Line 312 – Two/6 --- change to 2/6
v) Line 349 – Weltman, Fletcher and Rogers 2014 – correct to Weltman et al, 2014
vi) Line 351 – in vitro – write in italics in vitro
vii) Line 352 – Salazar 2024 – missing in references list
viii) Line 485 – 488 – Diaz et al 2020 – Missing in text of manuscript
ix) Line 551 – 552 – Swindle 2007 – Missing in text of the manuscript
x) Table 3 – In reference cited Newman ALW 2014 – remove initials
xi) Figure 1 – Write correct table number (Table 2 instead of Table 1)
f. Line 358, 359 – minor crossmatch between porcine red blood cells and bovine plasma was always incompatible in our study ………. Clarify!! Hemolysis and agglutination???
g. It is suggested that few images of Quick slide test & SSA may be included.
h. Table 2 – mention Grade 5 and Grade 6 – that has been depicted in the Figure 1
i. Grading in Figure 1 and Table 2 do not match.

Experimental design

a. Experiment is well planned. It has ample field importance in day-to-day veterinary practice. The investigation and methods employed are of high technical standards.
b. The research does fall within the scope of PeerJ

Validity of the findings

a. The conclusions are well drawn and supports the findings reported.

Additional comments

a. Plagiarism Checker – X reports 10 % similarity in the content of the manuscript, which is acceptable.
b. The following review article may also be referred to and used
Aravindh S and Ninan Jacob (2021). Blood transfusion in animals: A review. Journal of Entomology and Zoology Studies 9(5):357-361

---

## Round 0.2 · Minor Revisions

Dear authors,
We appreciate your efforts for making desired changes. But still few points are to be addressed. Please do the needful and resubmit asap. All the best

Reviewer 3 ·

Basic reporting

Thank you for adopting (and juggling) four different reviewers' proposed revisions. The manuscript's clarity and conclusions are much improved, and the authors have satisfactorily addressed all my previous comments. In addition to my answers to the authors' questions, I only have two minor revisions remaining.

Experimental design

I suggest calculating relative risk in addition to the reported p-value because it would provide readers with a better sense of the magnitude of how much more likely a porcine patient with type "A" blood is to be compatible with a bWB xenotransfusion than a porcine patient with type "0" blood.

To calculate the relative risk of compatibility for bovine WB with porcine blood type "A" or "0," use the following equation:

Relative risk = (% CMMa of bWB with porcine blood type "A") / (% CMMa of bWB with porcine blood type "0")

For example, if 80% of major crossmatches between bWB and porcine blood type "A" were compatible and 40% of major crossmatches between bWB and porcine blood type "0" were compatible, the relative risk = 80%/20% = 4. Thus, you could say in your Results that a xenotransfusion with bWB to a porcine patient with type "A" blood would be 4 times more likely to be compatible than a porcine patient with type "0" blood, which better communicates the clinical significance than a statistically significant p-value.

Validity of the findings

See below

Additional comments

Line 387-389: I think it would be appropriate for the authors to suggest a possible biological or evolutionary explanation in their Discussion for why a xenotransfusion from a bovine donor would be more compatible than a canine donor. Cows and domestic pigs fall within the same taxonomic order (Artiodactyla), versus dogs in the order Carnivora, so I think it would be fair to speculate their RBCs would be more similar or evolved from a more recent common ancestor's RBCs.

Table 3: Please consider changing "referenced image" to "Figure 1" since it's unclear from the table caption alone what image is referenced.

Figure 1: Please consider indicating that hemolysis increases from left to right, with 1 representing slight hemolysis and 6 representing marked/complete hemolysis

·

Basic reporting

I) The following points may be addressed
i) Line 83 – cite as Aravind and Ninan 2021
ii) Line 96 – change to Roux et al 2007a, 2007b
iii) Line 118, 368-369 – cite as Le Gal et al 2020
iv) Line 118, 173, 174, 234, 222, 253, 358, 389 – remove initials of author while citing in text
v) Line 218 – remove initials and cite as Hampton et al 2023a and Hampton et al 2023b
vi) Line 222 – cite as Hampton et al 2023b
vii) Line 343, 396 – Weltman, Fletcher and Rogers 2014 – cite as Weltman et al 2014
viii) Table 3 –Newman ALW 2014 – remove initials – cite as Newman 2014
ix) Provide six to seven Key words after the Abstract

Experimental design

Well Planned

Validity of the findings

Work is well planned with tables and figures supporting the findings.
Findings hold great value in Blood transfusion in the field and organized farms.

Additional comments

II) The corrections cited in the initial review have been incorporated
i) Images of Quick slide test & SSA have been included, as was suggested in the initial review
ii) Table 3 has been included describing the detectable hemolysis in the tubes
iii) Grading in Figure 1 and Table 3 match
iv) In Figure 1 the Tube 0 - indicating Grade 0 (as per Table 3) with no visually detectable hemolysis may be incorporated to compare the hemolysis in other tubes
v) Lines 359 – 377 clarify the doubts raised about minor crossmatch between porcine red blood cells and bovine plasma

---

## Round 0.3 · accepted · Accept

Dear authors,

In view ofthe recommendation of our esteemed reviewers, I am happy to inform you that the manuscript is accepted for publication in PeerJ. Please remember as it is only the editorial acceptance, you have to complete some production tasks before the final publication of your paper. Therefore, I advise you to be available for a few days to avoid any delays.

Good luck for your future submissions.

·

Basic reporting

All suggestions have been included and the manuscript is god for publication.

Experimental design

All suggestions have been included

Validity of the findings

All suggestions have been included

Reviewer 3 ·

Basic reporting

Thank you for your continued perseverance with this submission. The authors have satisfactorily addressed all my previous comments, and I have no further suggested edits or comments.

Experimental design

No comment

Validity of the findings

No comment

Additional comments

No comment

·

Basic reporting

Already reported in initials reviews

Experimental design

Already discussed in initials reviews

Validity of the findings

Already pointed out in initials reviews

Additional comments

The revisions have been made as per the reviewer suggestions.

In my view the article can be accepted for publication without any further revisions.